# Multi-Validity Process and Factor-Invariance. Perceived Self-Efficacy-Scale for the Prevention of Obesity in Preteens

**DOI:** 10.3390/children8060504

**Published:** 2021-06-14

**Authors:** Gilda Gómez-Peresmitré, Romana Silvia Platas Acevedo, Gisela Pineda-García, Rebeca Guzmán-Saldaña, Rodrigo Cesar León-Hernández, Nazira Calleja

**Affiliations:** 1Faculty of Psychology, National Autonomous University of Mexico, Av. Universidad 3004 Col Copilco-Universidad, Alcaldía Coyoacán, Mexico City C.P. 04510, Mexico; romsip@unam.mx (R.S.P.A.); ncalleja@unam.mx (N.C.); 2Faculty of Medicine and Psychology, Autonomous University of Baja California, Cal. University 14418, International Industrial Park, Tijuana C.P. 22390, Mexico; gispineda@uabc.edu.mx; 3Institute of Health Sciences, Autonomous University of the State of Hidalgo, Camino a Tilcuautla s/n Pueblo San Juan Tilcuautla, Hidalgo C.P. 42160, Mexico; rguzman@uaeh.edu.mx; 4National Council of Science and Technology. Av. Insurgentes Sur 1582, Col. Crédito Constructor. Alcaldía Benito Juárez, Mexico City C.P. 03940, Mexico; rcleonhe@conacyt.mx

**Keywords:** multi-validity process, factorial invariance, factorial model, self-efficacy, obesity prevention

## Abstract

Given the lack of scales with a robust psychometric assessment of self-efficacy related to obesity in early adolescence, we aimed to obtain an instrument with high-quality validity and reliability items. Nonrandom samples (*N* = 2371) classified boys (1174, *M* = 12.83, *SD* = 0.84) and girls (1197, *M* = 12.68, *SD* = 0.78) from Mexico City and some cities of the Mexican Republic with obesity rates near to the national level mean. A multi-validity process and structural invariance analysis using the Perceived Self-efficacy Scale for Obesity Prevention were performed. A two-factor—physical activity and healthy eating—model with high effect-sized values—girls R^2^ (0.88, *p* < 0.01) and boys R^2^ (0.87, *p* < 0.01)—were obtained. Each factor explained more than half of the variance with high-reliability coefficients in each group and acceptable adjustment rates. The self-efficacy scale proved to have only girls, an invariant factor structure, or a psychometric equivalence between the groups. The obtained scale showed that a two-factor structure is feasible and appropriate, according to the highest quality of validity and reliability.

## 1. Introduction

Obesity, an modern-day health problem, is present globally (in developed, developing, and undeveloped countries) and has been identified as the pandemic of our century [1]. In Mexico, obesity, understood as a problem of malnutrition, has become a public health condition [2]. The problem of malnutrition/obesity in emerging countries such as Mexico has become even more complex, because it persists as a food problem of poverty [3] and, at the same time, as an eating disorder (Anorexia Nervosa/Binge Eating) called “of abundance”. These last eating disorders are qualitatively different from the traditional ones derived from poverty and extreme poverty [3,4,5,6]; however, they represent the same high cost for the socioeconomic development of the countries that suffer from them.

In Mexico, the data from the National Survey of Health and Nutrition [7] indicated a combined prevalence of obesity (14.6%) and overweightness (23.8%) at the national level for adolescents between 12 and 19 years old of 38.4%. The rural population rates were 34.6% (21% overweight and 13.6% obese), while, for the urban population, they were 39.7% (24.7% overweight and 15% obese). Concerning the overweight rates, the National Health Nutritional Survey (ENSANUT) report (2012–2018) showed, by sex, that women increased 3.3% (from 23.7% to 27%), while, for obesity, they increased by 2% (12.1–14.1%). For men, overweightness increased by 1.1%, from 19.6% in 2012 to 20.7% in 2018. For obesity, the increase was 6%, from 14.5% to 15.1%.

ENSANUT [7] points out that, in the Mexican Republic, 14.6% suffer from obesity. For the state of Baja California Norte, it presented 18.3%, Hidalgo 13.2%, Tampico 16.9%, and CDMX 12.2%, and unlike the 2012 survey [8], there was no data presented by the combined prevalence of overweightness and obesity.

Self-efficacy (SE) has attracted the attention of researchers for its potentially influential role in health assessment and instrumentation. Recently, interest has been sparked in searching for the psychometric properties of measurement instruments in elder and adult populations [9,10]. However, research in this regard for the adolescent population is even more critical, because SE promotes attitudes and behaviors related to physical and psychological health. Some authors [11] have noted that SE, among other psychological factors, must be included as part of any preventive and intervention activity that aims to control body weight. In this regard, the Inventory of Perceived Self-Efficacy for Weight Control of Román et al. [12] was adapted [13] for school populations.

They developed a parsimonious final instrument (20 items) with acceptable internal validity and consistency (*α* = 0.88). This instrument allows measuring three primary components of self-efficacy relative to the prevention of overweightness: physical activity, risk eating, and control over emotional feeding. In SE research, the SE/health relationship stands out for the facilitating role it plays in achieving: (1) positive and long-lasting responses over the difficulties imposed by changes in harmful habits and (2) the relationship between a perception of high SE and an increase in security itself to solve problems [14,15].

Other researchers have reported that SE is particularly important as a predictor for the initial success and maintenance of weight loss, physical activity, and healthy eating [16,17,18]. According to McCabe [19], SE is formed by a subject’s social and cognitive experiences and their attentions and interpretations.

In a study with 279 adolescents, it was confirmed that beliefs related to SE and intentions of healthy eating are significant predictors of behaviors related to eating and a healthy weight [20]. Likewise, the importance of physical practice on the level of both general and specific SE has been documented. In a study on a random sample of 2079 adolescent boys and girls between the ages of 14 and 17, significant differences were found in the general SE scale among those who performed sport-based physical activity vs. those who did not [21]. Significant differences were found among those who practiced more frequently (weekly) vs. those who practiced less frequently. Recent studies have shown [22] that, during adolescence, physical activity decreases and sedentary leisure activities (e.g., electronic games, among others) increase.

The psychometric properties of the self-efficacy for healthy eating and physical activity measures (SE-HEPA) for preteens were examined [23]. The researchers reported α = 0.88 for the full scale, α = 0.85 for the healthy eating subscale, and α = 0.81 for the physical activity subscale. For the results of the exploratory factor analysis (EFA) and confirmatory factor analysis (CFA), the authors indicated an adequate construct validity using a two-factor model with an excellent global fit. They also reported good discriminant validity between the factors (0.51) and a correlation between the factors and an inverse correlation where higher body mass index (BMI) percentile scores were correlated with lower SE-HEPA scores.

The importance of perceived SE and its relationship with different behaviors, attitudes, and intentions associated with physical and mental health requires instruments with validity measurements in their broadest expressions that can ensure theoretical/practical congruence with reality. However, the SE measurements have shown some limitations, and the development of a psychometric assessment has been even more insufficient.

Our study is based on social cognitive theory [24], which has been widely adopted and explained precisely in obesity research in preadolescent and adolescent populations associated with the self-efficacy construction. Based on the indicated background and the lack of scales featuring robust psychometric assessments of SE related to health problems, especially for young people, we designed this study to obtain as the central objective an instrument with optimal psychometric properties for a population in early adolescence. To do this, we adapted and assessed the self-efficacy inventory for weight control [25] to obtain a multi-validated scale for obesity prevention (PSOP). We can define the process of multi-validity, as its name implies, as a joint analysis that addresses the psychometric objective of validity from different statistical characteristics that are required for the items of the instrument under study to comply with better scanning and measurements.

The principle of all research that works with constructs and seeks to compare them across groups is to ensure, with a view toward obtaining results with high validity scores, that they have the same meaning. This principle of semantic equality at a basic level can be covered by cognitive laboratories. At a more complex level of analysis, such as factorial invariance, the restriction of equality of the parameters between groups is increased at each step to prove that there is invariance. The highest level of restriction (strongest level of measurement) is structural invariance. In this way, in the present study, it was decided that the content, construct, and external validity would yield an accurate measurement of health-related self-efficacy, but ensuring an equivalent comparison (same meaning) between the groups through a factorial invariance would further refine the measurements. Thus, a culminating point of the present study was to ensure that the items of the instrument achieved structural invariance.

The following working hypotheses were formulated:

**Hypothesis** **1** **(H1).**
*The results of the EFA (exploratory factor analysis) will yield a two-dimensional structure of the instrument under study (PSOP), and the CFA (Confirmatory Factor Analysis) will confirm it.*


**Hypothesis** **2** **(H2).**
*Comparison between the groups (residence by sex) will demonstrate the structural invariance, which ensured the same meaning of the construct (self-efficacy factors).*


## 2. Methods

### 2.1. Design

The design of this study was an instrumental 2 × 2 × 2 (factors by sex by place of residence), with nonrandom samples and independent measurements.

### 2.2. Participants

The total sample (*N* = 2371) for nonrandom selection was formed by two independent samples of boys (1174) and girls (1197) with ages between 11 and 14 years from their first two years of high school. These participants were then subdivided for statistical purposes: exploratory and confirmatory factor analyses (EFA and CFA) at *n1* = EFA-Boys = 586 (*Mage* = 12.84, *SD* = 0.85), *n2* = CFA-Boys = 588 (*Mage* = 12.82, *SD* = 0.83), *n3* = EFA-Girls = 614 (*Mage* = 12.68, *SD = 0*.80), and *n4* = CFA-Girls = 583 (*Mage* = 12.68, *SD* = 0.77) (Table A1). Adolescents comprised each subsample from the four following cities of the Mexican Republic (MR, *N* = 2371): Mexico City (CMX, *n* = 864), Tijuana Baja California (TBC, *n* = 298), Pachuca Hidalgo (PH, *n* = 509), and Tampico Tamaulipas (TT, *n* = 700). For this study, an initial comparison was made by grouping the students from CMX with those from the other three cities in the MR. Subsequently, new comparisons between the CsMR and each one of these vs. CMX will be reviewed. One of the underlying reasons for the first comparison was to find out whether the responses to the instrument were made by boys and girls are influenced by their place of residence (a metropolis vs. a smaller city of the country).

### 2.3. Instrument

The self-efficacy inventory for weight control [25] was adapted for this study. This inventory with 40 items was validated with overweight and obese adolescents from the state of PH. These inventory items were adapted by agreement of the language and meaning among younger adolescent students 11–14 years old from the cities under study using the cognitive laboratories method control [26]. The content validity index with eight judges was CVI = 0.86. It covered the expected criteria (IVC ≥ 0.75 with ≤ 8 judges) [27]. The final instrument for boys was composed of 13 items with *α* = 0.90 and CRC = 0.93 distributed between 2 factors: self-efficacy physical activity (SPA) (α = 0.89 and CRC = 0.89) and self-efficacy towards healthy eating (SHE) (α = 0.85 and CRC = 0.85). For girls, the instrument grouped 14 items (α = 0.91 and CRC = 0.93) distributed along the same factors: SPA with α = 0.90, CRC = 0.89 and SHE with α = 0.86 and a CRC of 0.85. The item grouped only in the girls’ instruments was going up and down stairs instead of using an elevator or escalator. The participants responded to each item using a 4-point Likert scale that included response options ranging from “I am not able to do it” (1) to “I am very capable of doing it” (4). Higher scores corresponded with a higher level of perceived self-efficacy in overcoming barriers and increasing SPA and SHE behaviors. Girls and boys were excluded if they had a physical/medical health condition that would prevent them from engaging in physical activity or making decisions about food.

### 2.4. Procedure

The authorities of two public secondary schools from each of the target states under study were visited to request the students’ participation. Students were given an informed consent form to be signed by their parents or other responsible parties. Students whose parents consented completed the questionnaires applied by psychologists for approximately 30 min during their regular class time. The objectives of the research were explained to the students, and their voluntary participation was made explicit. Approval was obtained from the Ethics Committee of the National Autonomous University of Mexico and from the other institutions of the researchers who participated in the research. The students presented no rejections.

### 2.5. Statistical Analysis

The statistical package SPSS V. 22 (IBM, Mexico, Mexico) was used. The responses’ variability and normality were tested with a frequency analysis [28]. The multivariate normality was tested with the Mardia coefficient (<70) [29]. A *t*-test was used for the independent samples to determine the differences by sex. Before the factor analysis, compliance with the established criteria Kaiser- Meyer -Olkin (KMO) ≥ 70 was evaluated (Bartlett χ^2^ sphericity test *p* ≤ 0.05) [30]. The EFA was applied to samples 1 and 3 using the likelihood and Promax rotation (factor loads ≥ 0.40 and number of items ≥ 3 for each factor). The CFA was applied to samples 3 and 4 using AMOS 21 software [31]. The following goodness of fit indices were considered—χ^2^/df ≤ 3, Root Mean Square Error of Approximation (RMSEA ≤ 0.05), Goodness-of-Fit Index (GFI ≥ 0.90), Adjusted Goodness-of-Fit Index (AGFI ≥ 0.90), Comparative Fit Index (CFI ≥ 0.95), Non-Normed Fit Index (NNFI ≥ 0.90), and Standardized Root Mean Square Residual (SMRR ≤ 0.05) [32,33]. Comparisons of the models to test the factorial invariance (FI) were evaluated with Δχ2 > 0.05, ΔCFI ≤ 0.01, and ΔRMSEA ≤ 0.015. In the IF analysis, the set-up restraints method was used. Finally, for the reliability of the resulting means, Cronbach’s alpha (≥0.70, *p* ≤ 0.05) and the compound reliability coefficient (CRC) ≥ 0.70 [34] were used.

## 3. Results

### 3.1. Exploratory Factor Analysis EFA

Since the *t*-test showed significant differences between boys’ and girls’ responses to the PSOP, the EFA was applied to each boy (*n* = 586) and girl (*n* = 614) after analyzing their relevance using the KMO value and the significant Bartlett sphericity test. The results for both samples yielded two factors (16 items): F1—self-efficacy for the performance of the physical activity and F2—self-efficacy for healthy eating (SPA and SHE) that explained 49% (boys) and 48% (girls) of the variance.

### 3.2. Confirmatory Factor Analysis (CFA) and the Factorial Model (FM)

The CFA showed the same two factors as the EFA: SPA and SHE. For girls, the variance explained was 51% (14 items, factorial loads 0.60–0.80, α = 0.91 and CRC = 0.93; factor 1 (SPA) grouped seven items, α = 0.90 and CRC = 0.89; factor 2 (SHE) was formed by seven items, α = 0.86 and CRC = 0.85). For boys, the variance explained was 49% (13 items, factorial loads 0.58–0.84, general alpha = 0.90 and CRC = 0.93; factor 1, α = 0.89 and CRC = 0.89; factor 2, α = 0.85 and CRC = 0.85).

The two factors produced by the CFA maintained a correlation (0.61 and 0.66) for boys and girls and were subjected to a technique of modeling structural equations. The confirmatory model by sex was developed once the multivariate distribution of the data was proven (Mardia < 0.70; for girls = 37.63 and for boys = 30.87). For each item, Figure A1 and Figure A2 show mostly strong factorial loads, a high level of explained variance, and good values for the indexes of adjustment of the factors by sex.

### 3.3. External Validity and Convergent and Discriminant Validities

For boys and girls, the convergent validity results showed statistically significant factor loads exceeding the value of 0.50, with *p <* 0.05 for the standardized loads and 1.96 for the critical coefficients [35]. A high composite reliability coefficient was found for the boys’ and girls’ means, i.e., CRC values of 0.85–0.90. In Table A2, these values are above 0.70, which is the minimum required value. Additionally, they show that the self-efficacy for physical activity and self-efficacy towards healthy eating among boys and girls have good values. The results in Table A3 show compliance with this criterion—correlations between the factors must be lower than the square root of the average extracted variance—therefore, the discriminant validity is met.

## 4. Discussion

The current investigation’s final objective was to develop a multi-validated instrument with optimal psychometric properties with an invariant factor structure between the study groups for the prevention of obesity in early adolescence.

Two working hypotheses were formulated regarding the structure of the instrument (PSOP dimensions) and the level of measurement (structural invariance) of the comparison between the groups (residence by sex).

### 4.1. Content and Construct Validity

As already described, a first step was developed and founded an adequate content validity index to address the health aspects—physical activity/healthy eating. Moreover, through cognitive laboratories, we proved that the perceived self-efficacy scale items had the same meanings for all the participants. Related to the PSOP factorial structure, evidence was found that the EFA showed two factors. We cross-validated the EFA results via the CFA with independent new samples of boys and girls, and their results allowed us to confirm the evidence from the two-dimensional model provided by the EFA. These two factors were the same and retained the same order for boys and girls. In this way, H1 aimed at testing the bi-dimensionality of the instrument was accepted.

The same items were associated with the same factors in each group, showing stability throughout the analyses. Each factor explained more than half of the variance with the high reliability of the scores (alpha and CRC coefficients) in each group, presenting similar medium intra-correlation values. These factors obtained high and adequate adjustment rates (Table A2). These results for the factors related to healthy eating and physical activity and the bifactorial structure were consistent with those reported for the SE-HEPA factors [23]. However, this study’s instrument (i.e., the PSOP) had higher levels of psychometric properties, as described and shown throughout this report.

### 4.2. External Validity

In order to ensure that the same construct (e.g., self-efficacy) was measured for the different groups (e.g., CMX vs. CsMR), other criteria besides the measurement invariance, such as the evidence of convergent and discriminant validity, were required [35]. The first was achieved when the correlation between the variables of the same constructs was high and greater (convergent validity; CV) than the correlation between the variables of other constructs (discriminant validity; DV). Thus, the PSOP covered these evidence criteria extensively (Table A3). In the second, discriminant validity, the AVE value of each factor was higher than the square of the correlations between the construct/latent variables. However, because this variance was shared between constructs, mere comparative criterion (greater diagonal value than the value outside the diagonal) was necessary but not sufficient. This criterion was reinforced by the ability of the correlation to better discriminate the validity and lower the value correlation, as demonstrated in this study (Table A3). The DV was also met.

### 4.3. Factorial Invariance (FI)

As already noted, the measurement of structural invariance using instruments that measure self-efficacy in early adolescence has been poorly addressed. Few studies have demonstrated support for the results obtained at this level of measurement invariance for self-efficacy instruments [36]. Moreover, the measurement of invariance is a prerequisite for comparisons between groups. These measures must have the same meanings across the items of the factors SPA and SHE and mean the same for both the children of Mexico City (CMX) and for those from the other cities of the Mexican Republic (CsMR) to avoid, for example, biases in the interpretations of the results. Thus, we tested the results of the FI using the five levels of measurement. At the configurational invariance level, we confirmed that the PSOP was composed of two factors, with the same items loaded on the same factors regardless of the sex of the students. For girls, comparisons of the models at all levels of measurement, from the weakest to the most strict and structural, showed the models to be invariant (Table A4). Among girls, the PSOP measured the same construct. This strict invariance proved that the variance/covariance of the residual errors of the items (item uniqueness) was the same for the two groups; at the same time, this invariance proved the variance of the errors from the group differences in the common factors. Thus, the working hypothesis H2 was partially confirmed. Only the comparison between the groups of girls showed structural invariance (Table A4).

For boys (Table A5), the FI’s measurements were imperfect and presented some problems, e.g., in comparison the of M2, the invariances for the factor loads were rejected. Thus, although the comparisons of the strict and structural measurements of the residual and factor invariances were not significant (*p* = 0.056) (i.e., the criterion to be accepted as invariant), the non-invariance of M2 prevented the acceptance of this result [32].

The differences in the size of the effect could explain the differences between boys/girls, in which girls had higher overall self-efficacy and better self-perception in managing physical activity and healthy eating than boys. It may be that these psychological conditions—absent in boys—provide cognitive stability in the girls’ responses, since high levels of self-efficacy have also been shown to be associated with increased and maintained healthy behaviors. The authors should discuss the results and how they can be interpreted from the perspective of previous studies. The findings and their implications should be discussed in the broadest context possible. Future research directions may also be highlighted.

Starting from the fact that the structural invariance of the items of the Perceived Self-Efficacy for Obesity Prevention Scale (PSOP) was tested among girls living in a big city vs. small cities, what proceeds is to extend the generalization of our results. The first step in this direction is to test that the PSOP items have the same meaning using the BMI (obesity vs. thinness) as a comparison variable between groups of girls. In the present study, the girls came from cities with rates equal to or higher than the national average for obesity. The proposed research would refine the understanding of the role of obesity by controlling for the BMI variable concerning perceived self-efficacy. PSOP items have the same meaning for obese girls as they do for thin girls.

Returning to the findings of our study as a whole will allow us to see the need to obtain (taking care to meet the psychometric requirements imposed) complete information on the efficiency of the instrument. That is, it is required to disaggregate the global information (self-efficacy towards physical activity and nutritious eating) provided by the comparison of boys and girls living in two different places: in a large metropolitan city (CMX, 9,209,944 inhabitants) vs. in smaller cities. Intercomparisons between nonmetropolitan cities would enrich the information provided by the PSOP concerning its psychometric properties in its function of providing information on the self-efficacy that children perceive to engage in physical activity and follow a healthy diet. It is also advisable to investigate variables specific to a large city that may influence the perceived self-efficacy favoring obesity. Finally, it is suggested to confirm the influence of large cities by selecting other metropolitan cities.

## 5. Conclusions

Our study’s final product was an instrument (PSOP) with proven validity data whose bifactorial structure was supported by substantive statistical models directly related to obesity prevention. Besides, we obtained the most important statistical quality of the PSOP through the factorial invariance method. We proved that the items of the PSOP had a structural invariance for the girls. In this way, we positively confirmed the research hypotheses, although H2 was only partially confirmed, implying an essential difference between girls/boys that must be considered a priority issue for future research. In sum, the overall results of the multi-validity process showed for boys and girls acceptable item values for the content, construct, and external validity and that obtained from the factorial invariance. Evidence of the high values of internal consistency and reliability should be added. In the same way, we found the expected adjustment indices values. We conclude that the bifactorial structure of the PSOP was feasible and appropriate according to the highest quality of validity and reliability.

### Limitations

The main and major limitations were derived from the self-reported nature of measuring the PSOP factors and the lack of random samples during data collection. Another limitation of this study was that it was designed to perform a single comparison (CMX vs. CsMR), lacking more fine-grained information. Further research should overcome this constraint.

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
