# Peer review of "Multi-Validity Process and Factor-Invariance. Perceived Self-Efficacy-Scale for the Prevention of Obesity in Preteens"

_children, 2021, doi:10.3390/children8060504_

Round 1
Reviewer 1 Report
Thank you for the opportunity to review this manuscript. Overall, the manuscript is interesting. However, there are some points I think the authors may consider in order to improve the manuscript. At the moment, the main ones are:
- Review the text for minor errors (e.g. “the obesity” or maybe “obesity”?; It covered the expected criteria ( (IVC ≥ .75 with ≤ 8 judges) [22].).
- There are no references in many places (e.g. The condition of obesity is most complicated in countries that are not classified as “developed”, since such countries may face the same problem due to different structural and simultaneous causes. In Mexico, it has not been able to overcome malnutrition due to poverty. This type of malnutrition persists alongside malnutrition that is not due to poverty but instead due to ideas, values, and socio-economic contexts outside and characteristic of countries in which malnutrition is associated with mental pathologies, such as anorexia nervosa (which is a multifactorial problem related to the canons of Western ideology and lifestyle, e.g., fashion and extreme thinness as an ideal of beauty). Thus, although obesity obeys multiple and diverse causes, its persistence and increase is the same regardless of the country's level of development.).
- The statistics symbols (e.g. M, SD, p) should be in italics.
- The description of the participants is not transparent. Maybe the authors could have used the table for this purpose?
- The figures are not entirely well prepared. The statistics are superimposed on the arrows.
- Table A6 is not transparent (note the model description and the proportions between the columns, among other things).
- To a small extent the authors make suggestions as to how the results of their study need to be extended in the future to learn more about the issue in question.
- I would recommend expanding the Discussion section as to how some of the results of the study could be implemented rather than a repeat of what was already discussed.
- Discuss more limitations of this study.
Author Response
Comments and Suggestions for Authors
Thank you for the opportunity to review this manuscript. Overall, the manuscript is interesting. However, there are some points I think the authors may consider in order to improve the manuscript. At the moment, the main ones are:
1.Review the text for minor errors (e.g. “the obesity” or maybe “obesity”?; It covered the expected criteria ( (IVC ≥.75 with ≤8 judges) [22].).
A general review was made and errors were corrected(See words withred color)
2.There are no references in many places (e.g. The condition of obesity is most complicated in countries that are not classified as “developed”, since such countries may face the same problem due to different structural and simultaneous causes. In Mexico, it has not been able to overcome malnutrition due to poverty. This type of malnutrition persists alongside malnutrition that is not due to poverty but instead due to ideas, values, and socio-economic contexts outside and characteristic of countries in which malnutrition is associated with mental pathologies, such as anorexia nervosa (which is a multifactorial problem related to the canons of Western ideology and lifestyle, e.g., fashion and extreme thinness as an ideal of beauty). Thus, although obesity obeys multiple and diverse causes, its persistence and increase is the same regardless of the country's level of development.).
Dear reviewer 1, The first author of this manuscript is the author of this text based in her experience in this field.
3.The statistics symbols (e.g.M,SD,p) should be in italics.
All statistical symbols have been changed to italics.
4.The description of the participants is not transparent. Maybe the authors could have used the table for this purpose?
Added Table 1
5.The figures are not entirely well prepared. The statistics are superimposed on the arrows.
The figures have been fixed
6.Table A6 is not transparent (note the model description and the proportions between the columns, among other things).
The table A6 now Table 7 has been fixed
7.To a small extent the authors make suggestions as to how the results of their study need to be extended in the future to learn more about the issue in question.
The following content was added to the text:
Returning to the findings of our study as a whole will allow us to see the need to obtain (taking care to meet the psychometric requirements imposed) complete information on the efficiency of the instrument. That is, it is required to disaggregate the global information (self-efficacy towards physical activity and nutritious eating) provided by the comparison of boys and girls living in two different places: in a large metropolitan city (MCX, 9 209 944 inhabitants) vs. in smaller cities. Inter-comparisons between non-metropolitan cities would enrich the information provided by the PSOP -concerning its psychometric properties in its function of providing information on the self-efficacy that children perceive to engage in physical activity and follow a healthy diet. It is also advisable to investigate variables specific to the large city that may influence perceived self-efficacy favoring obesity. Finally, it is suggestedconfirming the influence of large cities by selectingother metropolitan cities.
8.I would recommend expanding the Discussion section as to how some of the results of the study could be implemented rather than a repeat of what was already discussed.
The following content was added to the text:
Besides, we obtained the most important statistical quality of the PSOP, through the factorial invariance method. We proved that the items of the PSOP have a structural invariance for the girls. In this way, we positively confirmed the research hypotheses, althoughH2was only partially confirmed, implying an essential difference between girls/boys that must be considered a priority issue for future research. In sum, the overall result of the multi-validity process showed for boys and girls acceptable item values for content, construct, and external validity and that obtained from the factorial invariance.
9.Discuss more limitations of this study.
Another limitation of this study is that it was designed to perform a single comparison (CMX vs. CsMR), losing more fine-grained information. Further research should overcome this constraint
Reviewer 2 Report
The work entitled “Multi-validity process and factor-invariance. Perceived self-ef- 2 ficacy-scale for the prevention of obesity in preteens.” contains new scientific knowledge and covers a relevant topic. However, I have some major comments that have to be addressed before it can be considered for publication.
In the introduction, I would suggest modifying the objectives section. I think that it is a bit odd as it appears right now. Also, I would introduce hypotheses for the study.
Authors should explain the reason for this: “Subsequently, for comparison pur- 136 poses, the students were groups as those from Mexico City vs. those from the three re- 137 maining cities of the MR” It seems that the decission is based only in the number of participants from each city.
Please check for gramatical errors: “…fit indices was considered”
I do not know if I fully understand the rationale for the analyses. Authors mention that a t test to check for gender differences was applied. However, they also explain that the ítems were different for boys and girls, and talk about two different samples of boys and girls.
Also, it appears that authors decide to compute the EFA separately for girls and boy attending to the t-test differences in PSOP scores. The fact that scores in X dimensions of a test are different does not imply that the dimensional structure is not equivalent.
Also, the measurement invariance (MI) (which is apparently conductes in the study) es not previously explained, nor is the mentioned the rationale for that. And also, the MI is attending to a variable, buty it is not possible to understand to what variable in this case, as the samples were already splitted by sex (I think that is better to talk about gender).
Finally, authors should consider revising the writing when talking about the study of internal consistency or convergent validity in some parts of the paper. First of all,
validity is not a property of the test but inferences of the scores, and also, there are sources or validity evidences, as it is reflected in the APA standards. Attending to this approach it would be more appropriate to talk about evidences of internal structure or evidences of relation with other variables or external variables. In addition, the reliability is not a characteristic of the test. It is more correct to talk about reliability of the scores or estimation of the reliability of the scores (Prieto & Delgado, 2010).
Author Response
Comments and Suggestions for Authors
The work entitled“Multi-validity process and factor-invariance. Perceived self-ef-2 ficacy-scale for the prevention of obesityin preteens.”contains new scientific knowledge and covers a relevant topic. However, I have some major comments that have to be addressed before it can be considered for publication.
1.In the introduction, I would suggest modifying the objectives section. I think that it is a bit odd as it appears right now. Also, I would introduce hypotheses for the study.
Objectives were changed to hypotheses
2.Authors should explain the reason for this: “Subsequently, for comparisonpurposes,136 the students were groups as those from Mexico City vs. those from the three remaining cities of the MR”137It seems that the decission is based only in the number of participants from each city.
Dear reviewer 2: In the abstract (lines 19-20) the reasons for the choice of the study cities were stated: Cities with obesity rates close to or higher than the national average of obesity.
According to your comment (section 2.2:136-138) the reasons for the comparison made (Mexico City vs. the three remaining cities) were added: (144-149).
For this study, an initial comparison was made by grouping students from CMXwith those from the other three cities in the MR. Subsequently, new comparisons between theCsMR and each one of these vs. CMXwill be reviewed. One of the underlying reasons for the first comparison is to find out whether responses to the instrument made by boys and girls are influenced by place of residence (a metropolis vs. a city of the country).
3. Please check for gramatical errors: “...fit indices was considered”.
The word was changed to were
4.I do not know if I fully understand the rationale for the analyses. Authors mention that a t test to check for gender differences was applied. However, they also explain that the ítems were different for boys and girls, and talk about two different samples of boys and girls.
5.Also, it appears that authors decide to compute the EFA separately for girls and boy attending to the t-test differences in PSOP scores. The fact that scores in X dimensions of a test are different does not imply that the dimensional structure is not equivalent.
Reviewer 2: responses to comments 4 and 5. We believe that we lacked information regarding the use of the t-test, so we would like to clarify and justify its use: Previous research data show that children from pre-adolescence onwards have different attitudes (depending on whether theyare male or female) towards risk factors related to Eating Disorders (e.g., body image, physical activity, eating behavior, obesity, and others). Hence, the t-test allowedconfirmation of such sex differences. Otherwise, we would work with a single samplesubjected to all the statistical and non-statistical procedures. The rationale for the use of samples for boys and girls was well justified by the valid estructural invariance results for girls but not for boys (self-efficacy factors have different meanings for boys than girls).
6.Also, the measurement invariance (MI) (which is apparently conductes in the study) es not previously explained, nor is the mentioned the rationale for that. And also, the MI is attending to a variable, buty it is not possible tounderstand to what variable in this case, as the samples were already splitted by sex (I think that is better to talk about gender).
Reviewer 2: In 20-25 we justify the use of the factorial invariance, as in rows 275to 290.You point out "thatthe measurement invariance (MI) is not previously explained". At all times, from the abstract,the introduction 1 (117-120); Statistical Analysis 2.5 (193); Discussion 4 (234-239) to the factorial invariance 4.3 (294) we make it explicit that the data will be subjected to a Factorial Invariance (FI) and that we seek to test whether the instrument (its items) has structural invariance. MI is the firstmetric level, constraining all factor loadings to be equal across the twoGroups,the second of the 5
levels of factorial invariance analysis. We obtained the MI but our ultimate purpose was level 5, the strongest level of measurement (structural invariance) all the measurement levels can be seen in Tables 5 and 6. All the measurements carried out with the invariance factor analysis were performed on the answers given to the two factors of the variable self-efficacy.
Regarding the proposal to use the word gender rather than sex, it seems to us very well, however, we would like to comment with you that within the social and behavioral sciences, the use of these terms has been differentiated. Sex as the merely biological characteristic of each individual; while gender has been left to talk about ethnic and socio-cultural differences due to socialization. If you have no objection we would like to make no changes.
7. Finally, authors should consider revising the writing when talking about the study of internal consistency or convergent validity in some parts of the paper. First of all,validity is not a property of the test but inferences of the scores, and also, there are sources or validity evidences, as it is reflected in the APA standards. Attending to this approach it would be more appropriate to talk about evidences of internal structure or evidences of relation with other variables or external variables. In addition, the reliability is not a characteristic of the test. It is more correct to talk about reliability of the scores or estimation of the reliability of the scores (Prieto & Delgado, 2010).
In relation to your comments (number 7) we understand and agree with them, we could summarize that in general the psychometric properties (such as validity and reliability) are not of the instruments, but of their items and results. In view of this, in different parts of the text the necessary arrangements were made as already indicated.We added the word evidence in the following lines: 4.1 (245, 247and 261,264); 4.2 (328).
Round 2
Reviewer 1 Report
Thank you for the opportunity to review a revised manuscript. I appreciate the authors' effort in responding to all comments, and I believe that some of the issues have been appropriately addressed. However, there are still issues that raise my concerns:
- I think that if authors present any information, they should be mainly supported in the literature (in my opinion it is problematic that such extensive parts of the text have no references). When reading parts of the text without references, I get the impression that after exploring the literature, the authors would find references supporting their experience in this field.
- Still some statistical symbols are not italicized, e.g. “SD” -> “Non-random sample (N = 2371) grouped boys (1174, M =12.83, SD =.84)…”
- In my opinion, the description of participants is still not transparent (more information should be included in the table).
- The authors replied "The table A6 now Table 7 has been fixed" and in the current version of the manuscript the last table is table A5.
- I believe the authors did not fully respond to this comment “I would recommend expanding the Discussion section as to how some of the results of the study could be implemented rather than a repeat of what was already discussed.”.
Reviewer 2 Report
Authors have addressed all my comments. I have no further comments.
Author Response
There were no comments or suggestions from the Reviewer